# Easy—Ensemble Augmented-Shot-Y-Shaped Learning: State-of-the-Art Few-Shot Classification with Simple Components

**DOI:** 10.3390/jimaging8070179

**Published:** 2022-06-24

**Authors:** Yassir Bendou, Yuqing Hu, Raphael Lafargue, Giulia Lioi, Bastien Pasdeloup, Stéphane Pateux, Vincent Gripon

**Affiliations:** 1IMT Atlantique, Technopole Brest Iroise, 29238 Brest, France; yuqing.hu@imt-atlantique.fr (Y.H.); raphael.lafargue@imt-atlantique.fr (R.L.); giulia.lioi@imt-atlantique.fr (G.L.); bastien.pasdeloup@imt-atlantique.fr (B.P.); 2Orange Labs, 35510 Rennes, France; stephane.pateux@orange.com

**Keywords:** few-shot learning, classification, deep learning, augmentations, self-supervision, ensembling, backbones, cropping, ambiguity

## Abstract

Few-shot classification aims at leveraging knowledge learned in a deep learning model, in order to obtain good classification performance on new problems, where only a few labeled samples per class are available. Recent years have seen a fair number of works in the field, each one introducing their own methodology. A frequent problem, though, is the use of suboptimally trained models as a first building block, leading to doubts about whether proposed approaches bring gains if applied to more sophisticated pretrained models. In this work, we propose a simple way to train such models, with the aim of reaching top performance on multiple standardized benchmarks in the field. This methodology offers a new baseline on which to propose (and fairly compare) new techniques or adapt existing ones.

## 1. Introduction

Learning with few examples, or few-shot learning, is a domain of research that has become increasingly popular in the past few years. Reconciling the remarkable performances of deep learning (DL), which are generally obtained thanks to access to huge databases, with the constraint of having a very small number of examples may seem paradoxical. Yet, the answer lies in the ability of DL to transfer knowledge acquired when solving a previous task toward a different, new one.

The classical few-shot setting consists of two parts:A base dataset, which contains many examples of many classes. Since this dataset is large enough, it can be used to efficiently train DL architectures. Authors often use the base dataset alongside a validation dataset. As is usual in classification, the base dataset is used during training, and the validation dataset is then used as a proxy to measure generalization performance on unseen data and, therefore, can be leveraged to optimize the hyperparameters. However, contrary to common classification settings, in few-shot, the validation and base datasets usually contain distinct classes, so that the generalization performance is assessed on new classes [1]. Learning good feature representations from the base dataset can be performed with multiple strategies, as will be further discussed in Section 2;A novel dataset, which consists of classes that are distinct from those of the base and validation datasets. We are only given a few labeled examples for each class, resulting in a few-shot problem. The labeled samples are often called the support set and the remaining ones the query set. When benchmarking, it is common to use a large novel dataset from which artificial few-shot tasks are sampled uniformly randomly, what we call a run. In that case, the number of classes *n* (named ways), the number of shots per class *k*, and the number of query samples per class *q* are given by the benchmark. This setting is referred to as *n*-way-*k*-shot learning. Reported performances are often averaged over a large number of runs.

In order to exploit knowledge previously learned by models on the base dataset, a common approach is to remove their final classification layer. The resulting models, now seen as feature extractors, are generally termed backbones and can be used to transform the support and query datasets into feature vectors. This is a form of transfer learning. In this work, we do not consider the use of additional data such as other datasets [2], nor semantic information [3]. Additional preprocessing steps may also be used on the samples and/or on the associated feature vectors, before the classification task. Another major approach uses meta-learning [4,5,6,7,8,9], as mentioned in Section 2.

It is important to distinguish two types of problems:In inductive few-shot classification, only the support dataset is available to the few-shot classifier, and prediction is performed on each sample of the query dataset independently of each other [9];In transductive few-shot classification, the few-shot classifier has access to both the support and the full query datasets when performing predictions [10].

Both problems have connections with real-world situations. In general, inductive few-shot corresponds to cases where data acquisition is expensive. This is the case for FMRI data, for example, where it is difficult to generalize from one patient to another and collect hours of training data on a patient could be harmful [11]. Alternatively, transductive few-shot corresponds to cases where data labeling is expensive. Such a situation can occur when experts must properly label data, but the data themselves are obtained cheaply, for instance in numerous medical applications [12,13].

In recent years, many contributions have introduced methodologies to cope with few-shot problems. There are many building blocks involved, including distillation [14], contrastive learning [15], episodic training [16], mixup [17], manifold mixup [1,18], and self-supervision [1]. As a consequence, it can appear quite opaque what the effective components are and whether their performance can be reproduced across different datasets or settings. Moreover, we noticed that many of these contributions report baseline performances that can be outperformed with a simpler training pipeline.

In this paper, we are interested in proposing a very simple method combining components commonly found in the literature and yet achieving competitive performance. We believe that this contribution will help have a clearer view on how to efficiently implement few-shot classification for real-world applications. Our main motivation is to define a new baseline with good hyperparameters and training routines to compare to and to start with, on which obtaining a performance boost will be much more challenging than starting from a poorly trained backbone. We also aim at showing that a simple approach reaches higher performance than increasingly complex methods proposed in the recent few-shot literature.

More precisely, in this paper:We introduce a very simple methodology, illustrated in Figure 1, for both inductive and transductive few-shot classification.We show the ability of the proposed methodology to reach or even beat state-of-the-art [9,19] performance on multiple standardized benchmarks of the field.All our models, obtained feature vectors, and training procedures are freely available online on our github: https://github.com/ybendou/easy accessed on 14 June 2022;We also propose a simple demonstration of our method using live video streaming to perform few-shot classification. The code is available at https://github.com/RafLaf/webcam accessed on 14 June 2022.

## 2. Related Work

There have been many approaches proposed recently in the field of few-shot classification. We introduce some of them following the classical pipeline. Note that our proposed methodology uses multiple building blocks from those presented hereafter.

### 2.1. Data Augmentation

First, data augmentation or augmented sampling are generally used on the base dataset to artificially produce additional samples, for example using rotations [1], crops [20], jitter, GANs [21,22], or other techniques [23]. Data augmentation on support and query sets, however, is less frequent. Approaches exploring this direction include [15], where the authors propose to select the foreground objects of images by identifying the right crops using a relatively complex mechanism, and [24], where the authors propose to mimic the neighboring base classes’ distribution to create augmented latent space vectors.

In addition, mixup [17] and manifold mixup [18] are also used to address the challenging lack of data. Both can be seen as regularization methods through linear interpolations of samples and labels. Mixup creates linear interpolations at the sample level, while manifold mixup focuses on feature vectors.

### 2.2. Backbone Training

Mixup is often used in conjunction with self-supervision (S2) [1] to make backbones more robust. Most of the time, S2 is implemented as an auxiliary loss meant to train the backbone to recognize which transformation was applied to an image [25].

A well-known training strategy is episodic training. The idea behind it boils down to having the same training and test conditions. Thus, the backbone training strategy, often based on gradient descent, does not select random batches, but uses batches designed as few-shot problems [4,16,26,27].

Meta-Learning, or learning to learn, is a major line of research in the field. This method typically learns a good initialization or a good optimizer such that new classes can be learned in a few gradient steps [4,5,6,7,8,9]. In this regard, episodic training is often used, and recent work leveraged this concept to generate augmented tasks in the training of the backbone [28].

Contrastive learning aims to train a model to learn to maximize similarities between transformed instances of the same image and minimize agreement between transformed instances of different images [15,29,30,31]. Supervised contrastive learning is a variant that has been recently used in few-shot classification, where similarity is maximized between instances of a class instead of the same image [14,32].

### 2.3. Exploiting Multiple Backbones

Distillation has been recently used in the few-shot literature. The idea is to transfer knowledge from a teacher model to a student model by forcing the latter to match the joint probability distribution of the teacher [14,33].

Ensembling consists of the concatenation of features extracted by different backbones. It was used to improve performances in few-shot classification [28]. It can be seen as a more straightforward alternative to distillation. To limit the computationally expensive training of multiple backbones, some authors propose the use of snapshots [34].

### 2.4. Few-Shot Classification

Over the past few years, classification methods in the inductive setting have mostly relied on simple methods such as nearest class mean [35], cosine classifiers [36], and logistic regression [24].

More diverse methods can be implemented in the transductive setting. Clustering algorithms [15], embedding propagation [37], and optimal transport [38] were leveraged successfully to outrun performances in the inductive setting by a large margin.

## 3. Methodology

The proposed methodology consists of 5 steps, described hereafter and illustrated in Figure 1. In the experiments, we also report ablation results when omitting the optional steps.

### 3.1. Backbone Training (Y)

We used data augmentation with random resized crops, random color jitters, and random horizontal flips, which is standard in the field.

We used a cosine-annealing scheduler [39], where at each step, the learning rate is updated. During a cosine cycle, the learning rate evolves between $\eta_{0}$ and 0. At the end of the cycle, we warm restart the learning procedure and start over with a diminished $\eta_{0}$. We start with $\eta_{0}=0.1$ and reduce $\eta_{0}$ by 10% at each cycle. We use 5 cycles with 100 epochs each.

We trained our backbones using the methodology called S2M2R described in [1]. Basically, the principle is to take a standard classification architecture (e.g., ResNet12 [40]) and branch a new logistic regression classifier after the penultimate layer, in addition to the one used to identify the classes of input samples, thus forming a Y-shaped model (cf. Figure 1). This new classifier is meant to retrieve which one of four possible rotations (quarters of 360° turns) has been applied to the input samples. We used a two-step forward–backward pass at each step, where a first batch of inputs is only fed to the first classifier, combined with manifold mixup [1,18]. A second batch of inputs is then has arbitrary rotations applied, and this is fed to both classifiers. After this training, the backbones are frozen.

We experimented using a standard ResNet12 as described in [40], where the feature vectors are of dimension 640. These feature vectors are obtained by computing a global average pooling over the output of the last convolution layer. Such a backbone contains ∼12 million trainable parameters. We also experimented with reduced-size ResNet12, denoted ResNet12$\left(\frac{1}{2}\right)$, where we divided each number of feature maps by 2, resulting in feature vectors of dimension 320, and ResNet12$\left(\frac{1}{\sqrt{2}}\right)$, where the number of feature maps are divided roughly by $\sqrt{2}$, resulting in feature vectors of dimension 450. The numbers of parameters are respectively ∼3 million and ∼6 million.

Using common notations of the field, if we denote x as an input sample and *f* as the mathematical function of the backbone, then z=f(x) denotes the feature vector associated with x.

From this point on, we used the frozen backbones to extract feature vectors from the base, validation, and novel datasets.

### 3.2. Augmented Samples

We propose to generate augmented feature vectors for each sample from the novel dataset. We did not perform this in the validation set as it is very computationally expensive. To this end, we used random resized crops from the corresponding images. We obtained multiple versions of each feature vector and averaged them. The literature has extensively studied the role of augmentations in deep learning [41]. Here, we assumed most crops would contain the object of interest. Therefore, the average feature vector can be used. On the other hand, color jitter might be an invalid augmentation since some classes rely extensively on their colors to be detected (e.g., birds or fruits).

In practice, we used ℓ=30 crops per image, as larger values do not benefit accuracy much. This step is optional.

### 3.3. Ensemble of Backbones

To boost performance even further, we propose to concatenate the feature vectors obtained from multiple backbones trained using the previously described method, but with different random seeds. To perform fair comparisons, when comparing a backbone with an ensemble of *b* backbones, we reduced the number of parameters per backbone such that the total number of parameters remains identical. We believe that this strategy is an alternative to performing distillation, with the interest of not requiring extra parameters and being a relatively straightforward approach. Again, this step is optional, and we perform ablation tests in the next section.

### 3.4. Feature Vector Preprocessing

Finally, we applied two transformations as in [35] on feature vectors z. Denote $\bar{z}$ the average feature vector of the base dataset if in the inductive setting or of the few-shot problem if in transductive setting. The ideal $\bar{z}$ would center the vectors of the few-shot runs around 0 and, therefore, would be the average vector of the combined support and query set. The number of samples being too small to compute a meaningful average vector in the inductive setting, we made use of the base dataset. In the transductive setting, queries are added to the support set for mean computation. The average vector is therefore less noisy and can be used to compute $\bar{z}$. The first operation (*C*—centering of z) consists of computing:(1)$$z_{C}=z-\bar{z}\;.$$

The second operation (*H*—projection of $z_{C}$ on the hypersphere) is then:(2)$$z_{CH}=\frac{z_{C}}{∥z_{C}{∥}_{2}}\;.$$

### 3.5. Classification

Let us denote $S_{i}$$\left(i\in \{1,\ldots ,n\}\right)$ the set of feature vectors (preprocessed as $z_{CH}$) corresponding to the support set for the *i*-th considered class and Q the set of (also preprocessed) query feature vectors.

In the case of inductive few-shot classification, we used a simple nearest class mean classifier (NCM). Predictions are obtained by first computing class barycenters from labeled samples:(3)$$\forall i:\bar{c_{i}}=\frac{1}{|S_{i}|}\sum_{z\in S_{i}}z\;,$$
then associating with each query the closest barycenter:(4)$$\forall z\in Q:C_{ind}\left(z,\left[\bar{c_{1}},\ldots ,\bar{c_{n}}\right]\right)=\arg\min_{i}{∥z-\bar{c_{i}}∥}_{2}\;.$$

In the case of transductive learning, we used a soft K-means algorithm. We computed the following sequence indexed by *t*, where the initial $\bar{c_{i}}$ are computed as in Equation (Equation 3):(5)$$\forall i,t:\left\{\begin{array}{ccc} {\bar{c_{i}}}^{0} & = & \bar{c_{i}}\;,\\ {\bar{c_{i}}}^{t+1} & = & \sum_{z\in S_{i}\cup Q}\frac{w(z,{\bar{c_{i}}}^{t})}{\sum_{z^{\prime }\in S_{i}\cup Q}w\left(z^{\prime },{\bar{c_{i}}}^{t}\right)}z\;, \end{array}\right.$$
where $w(z,{\bar{c_{i}}}^{t})$ is a weighting function on z, which gives it a probability of being associated with barycenter ${\bar{c_{i}}}^{t}$:(6)$$w\left(z,{\bar{c_{i}}}^{t}\right)=\left\{\begin{array}{ccc} \frac{\exp\left(-\beta {∥z-{\bar{c_{i}}}^{t}∥}_{2}^{2}\right)}{\sum_{j=1}^{n}\exp\left(-\beta {∥z-{\bar{c_{j}}}^{t}∥}_{2}^{2}\right)} & \mathrm{if} & z\in Q\;,\\ 1 & \mathrm{if} & z\in S_{i}\;. \end{array}\right.$$

Contrary to the simple K-means algorithm, we used a weighted average where weight values are calculated via a decreasing function of the $L_{2}$ distance between data points and class barycenters—here, a softmax adjusted by a temperature value β. In our experiments, we used β=5, which led to consistent results across datasets and backbones. In practice, we use a finite number of steps. By denoting $c_{i}^{\infty }$ the resulting vectors, the predictions are:(7)$$\forall z\in Q:C_{tra}\left(z,\left[{\bar{c_{1}}}^{\infty },\ldots ,{\bar{c_{n}}}^{\infty }\right]\right)=\arg\min_{i}{∥z-{\bar{c_{i}}}^{\infty }∥}_{2}\;.$$

## 4. Results

### 4.1. Ranking on Standard Benchmarks

We first report results comparing our method with the state-of-the-art using classical settings and datasets. We used the following datasets:MiniImagenet: A dataset extracted from ImageNet with 64 base classes, 16 validation classes, and 20 novel classes. Each class contains 600 images. The resolution is (84 × 84);TieredImagenet: Another subset of ImageNet with 351 base classes, 97 validation classes, and 160 novel classes. Classes contain a variable number of samples, usually about 1300. The resolution is (84 × 84);CUB-FS (Caltech-UCSD Birds-200-2011): This dataset is particularly challenging because it is only composed of pictures of birds. There are 100 base classes, 50 validation classes, and 50 novel classes. The number of images by class is not constant, close to 60. The resolution is (50 × 50);FC-100 (Fewshot-CIFAR-100): This is a subset of CIFAR 100 (Canadian Institute for Advanced Research 100). There are 60 base, 20 validation, and 20 novel classes containing 600 images. Images have a low resolution (32 × 32);CIFAR-FS (CIFAR-Fewshot): This is also a subset of CIFAR 100. There are 60 base, 16 validation, and 20 novel classes containing 600 images. Images have a low resolution (32 × 32).

For each method, we specified the number of trainable parameters and the accuracy of 1-shot or 5-shot runs. Experiments always used q=15 query samples per class, and results were averaged over 10,000 runs. Results are presented in Table 1, Table 2, Table 3, Table 4 and Table 5 for the inductive setting and Table 6, Table 7, Table 8, Table 9 and Table 10 for the transductive setting (the codes allowing for the reproduction of our experiments are available at https://github.com/ybendou/easy). Reported results for the existing methods are those specified by their respective papers. Some methods do not include their standard deviation over multiple runs.

Let us first emphasize that our proposed methodology shows a new state-of-the-art performance for MiniImageNet (inductive), TieredImageNet (inductive 1-shot setting) and FC100 (transductive), while showcasing competitive or overlapping results on other benchmarks. We believe that, combined with other more elaborate methods, these results could be improved by a fair margin, leading to a new standard of performance for few-shot benchmarks. In the transductive setting, the proposed methodology is less often ranked #1, but contrary to many alternatives, it does not use any prior on class balance in the generated few-shot problems. We provide such experiments in the Supplementary Material, where we show that the proposed method greatly outperforms existing techniques when considering imbalanced classes. Overall, our method has the benefit of being simpler while achieving competitive performance over multiple benchmarks.

### 4.2. Ablation Study

To better understand the relative contributions of components in the proposed method, we also compare, for each dataset, the performance of various combinations in Table 11 for the inductive setting and Table 12 for the transductive setting. Interestingly, the full proposed methodology (EASY) is not always the most efficient. We believe that for large datasets such as MiniImageNet and TieredImageNet, the considered ResNet12 backbones contain too few parameters. When reducing this number for ensemble solutions, the drop of performance due to the reduction in size is not compensated by the diversity of the multiple backbones. All things considered, only AS is consistently beneficial to the performance.

### 4.3. Discussion

Regarding the inductive setting, the proposed method achieves state-of-the-art performance on MiniImagenet by a fair margin. On TieredImagenet, only S2M2R performs better in the five-shot setting. This can be explained by the fact that TieredImagenet is the largest of the considered datasets and it requires more parameters to be trained efficiently, reducing the effectiveness of the proposed ensemble approach. We also noticed subpar performance on CIFAR-FS, and we believe that this is due to the small resolution of images in the dataset, which cripples the augmented sample step. On the FC-100 dataset, our results in the inductive setting overlap with [46]; however, our method has the advantage of having lower confidence intervals compared to other methods on the same benchmark. Regarding the transductive setting, our method achieves competitive results without any prior on the number of classes. This is important since multiple methods tend to fail when the number of samples per class is different, which we show in the Supplementary Material. Our explanation is that multiple methods tend to overexploit this prior. This concern was first raised by [57]. Overall, our method is easy to implement and requires few hyperparameters to be tuned compared to other competitive methods.

## 5. Conclusions

In this paper, we introduced a simple backbone to perform few-shot classification in both inductive and transductive settings. Combined with augmented samples and ensembling, we showed its ability to reach state-of-the-art results when deployed using simple classifiers on multiple standardized benchmarks, even beating previous methods by a fair margin (>1%) in some cases. We expect this methodology to serve as a baseline for future work.

## Figures and Tables

**Figure 1 jimaging-08-00179-f001:**
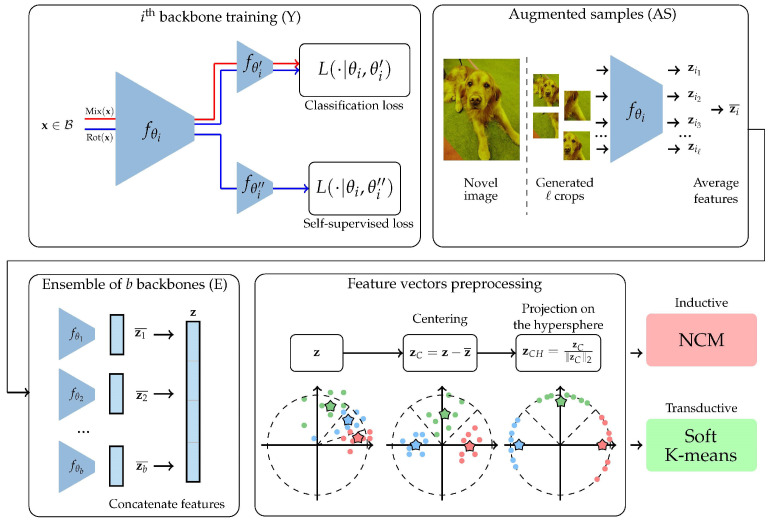
Illustration of our proposed method. Y: We first train multiple backbones using the base and validation datasets. We use two cross-entropy losses in parallel: one for the classification of base classes and the other for the self-supervised targets (rotations). We also use manifold mixup [18]. All the backbones are trained using the exact same routine, except that their initialization is different (random) and the order in which data batches are presented is also potentially different. AS: Then, for each image in the novel dataset and each backbone, we generate multiple crops, then compute their feature vectors, which we average. E: Each image becomes represented as the concatenation of the outputs of AS for each of the trained backbones. Preprocessing: We add a few classical preprocessing steps, including centering by removing the mean of the feature vectors of the base dataset in the inductive case, or the few-shot run feature vectors for the transductive case, and projecting on the hypersphere. Finally, we use a simple nearest class mean classifier (NCM) if in the inductive setting or a soft K-means algorithm in the transductive setting.

**Table 1 jimaging-08-00179-t001:** The 1-shot and 5-shot accuracy of state-of-the-art methods and the proposed solution on MiniImageNet in the inductive setting.

	Method	1-Shot	5-Shot
≤12 M c	SimpleShot [35]	62.85±0.20	80.02±0.14
	Baseline++ [36]	53.97±0.79	75.90±0.61
	TADAM [42]	58.50±0.30	76.70±0.30
	ProtoNet [16]	60.37±0.83	78.02±0.57
	R2-D2 (+ens) [28]	64.79±0.45	81.08±0.32
	FEAT [43]	66.78	82.05
	CNL [44]	67.96±0.98	83.36±0.51
	MELR [45]	67.40±0.43	83.40±0.28
	Deep EMD v2 [20]	68.77±0.29	84.13±0.53
	PAL [14]	69.37±0.64	84.40±0.44
	invariance-equivariance [46]	67.28±0.80	84.78±0.50
	CSEI [19]	68.94±0.28	85.07±0.50
	COSOC [15]	69.28±0.49	85.16±0.42
	EASY 2×ResNet12$\left(\frac{1}{\sqrt{2}}\right)$ (ours)	70.63±0.20	86.28±0.12
36 M c	S2M2R [1]	64.93±0.18	83.18±0.11
	LR + DC [24]	68.55±0.55	82.88±0.42
	EASY 3×ResNet12 (ours)	71.75±0.19	87.15±0.12

**Table 2 jimaging-08-00179-t002:** The 1-shot and 5-shot accuracy of state-of-the-art methods and the proposed solution on TieredImageNet in the inductive setting.

	Method	1-Shot	5-Shot
≤12 M c	SimpleShot [35]	69.09±0.22	84.58±0.16
	ProtoNet [16]	65.65±0.92	83.40±0.65
	FEAT [43]	70.80±0.23	84.79±0.16
	PAL [14]	72.25±0.72	86.95±0.47
	DeepEMD v2 [20]	74.29±0.32	86.98±0.60
	MELR [45]	72.14±0.51	87.01±0.35
	COSOC [15]	73.57±0.43	87.57±0.10
	CNL [44]	73.42±0.95	87.72±0.75
	invariance-equivariance [46]	72.21±0.90	87.08±0.58
	CSEI [19]	73.76±0.32	87.83±0.59
	ASY ResNet12 (ours)	74.31±0.22	87.86±0.15
36 M c	S2M2R [1]	73.71±0.22	88.52±0.14
	EASY 3×ResNet12 (ours)	74.71±0.22	88.33±0.14

**Table 3 jimaging-08-00179-t003:** The 1-shot and 5-shot accuracy of state-of-the-art methods and the proposed solution on **CIFAR-FS** in the **inductive** setting.

	Method	1-Shot	5-Shot
≤12 M c	S2M2R [1]	63.66±0.17	76.07±0.19
	R2-D2 (+ens) [28]	76.51±0.47	87.63±0.34
	invariance-equivariance [46]	77.87±0.85	89.74±0.57
	EASY 2×ResNet12$\left(\frac{1}{\sqrt{2}}\right)$ (ours)	75.24±0.20	88.38±0.14
36 M c	S2M2R [1]	74.81±0.19	87.47±0.13
	EASY 3×ResNet12 (ours)	76.20±0.20	89.00±0.14

**Table 4 jimaging-08-00179-t004:** The 1-shot and 5-shot accuracy of state-of-the-art methods and the proposed solution on CUB-FS in the inductive setting.

	Method	1-Shot	5-Shot
≤12 M c	FEAT [43]	68.87±0.22	82.90±0.10
	ProtoNet [16]	66.09±0.92	82.50±0.58
	DeepEMD v2 [20]	79.27±0.29	89.80±0.51
	EASY 4×ResNet12$\left(\frac{1}{2}\right)$ (ours)	77.97±0.20	91.59±0.10
36 M c	S2M2R [1]	80.68±0.81	90.85±0.44
	EASY 3×ResNet12 (ours)	78.56±0.19	91.93±0.10

**Table 5 jimaging-08-00179-t005:** The 1-shot and 5-shot accuracy of state-of-the-art methods and the proposed solution on FC-100 in the inductive setting.

	Method	1-Shot	5-Shot
≤12 M c	DeepEMD v2 [20]	46.60±0.26	63.22±0.71
	TADAM [42]	40.10±0.40	56.10±0.40
	ProtoNet [16]	41.54±0.76	57.08±0.76
	invariance-equivariance [46]	47.76±0.77	65.30±0.76
	R2-D2 (+ens) [28]	44.75±0.43	59.94±0.41
	EASY 2×ResNet12$\left(\frac{1}{\sqrt{2}}\right)$ (ours)	47.94±0.19	64.14±0.19
36 M	EASY 3×ResNet12 (ours)	48.07±0.19	64.74±0.19

**Table 6 jimaging-08-00179-t006:** The 1-shot and 5-shot accuracy of state-of-the-art methods and the proposed solution on MiniImageNet in the transductive setting.

	Method	1-Shot	5-Shot
≤12 M c	TIM-GD [47]	73.90	85.00
	ODC [48]	77.20±0.36	87.11±0.42
	PEM${}_{n}$E-BMS* [38]	80.56±0.27	87.98±0.14
	SSR [49]	68.10±0.60	76.90±0.40
	iLPC [50]	69.79±0.99	79.82±0.55
	EPNet [37]	66.50±0.89	81.60±0.60
	DPGN [51]	67.77±0.32	84.60±0.43
	ECKPN [52]	70.48±0.38	85.42±0.46
	Rot + KD + POODLE [53]	77.56	85.81
	EASY 2×ResNet12$\left(\frac{1}{\sqrt{2}}\right)$ (ours)	82.31±0.24	88.57±0.12
36 M c	SSR [49]	72.40±0.60	80.20±0.40
	fine-tuning(train+val) [54]	68.11±0.69	80.36±0.50
	SIB + E${}^{3}$BM [55]	71.40	81.20
	LR + DC [24]	68.57±0.55	82.88±0.42
	EPNet [37]	70.74±0.85	84.34±0.53
	TIM-GD [47]	77.80	87.40
	PT+MAP [56]	82.92±0.26	88.82±0.13
	iLPC [50]	83.05±0.79	88.82±0.42
	ODC [48]	80.64±0.34	89.39±0.39
	PEM${}_{n}$E-BMS* [38]	83.35±0.25	89.53±0.13
	EASY 3×ResNet12 (ours)	84.04±0.23	89.14±0.11

**Table 7 jimaging-08-00179-t007:** The 1-shot and 5-shot accuracy of state-of-the-art methods and the proposed solution on CUB-FS in the transductive setting.

	Method	1-Shot	5-Shot
≤12 M c	TIM-GD [47]	82.20	90.80
	ODC [48]	85.87	94.97
	DPGN [51]	75.71±0.47	91.48±0.33
	ECKPN [52]	77.43±0.54	92.21±0.41
	iLPC [50]	89.00±0.70	92.74±0.35
	Rot + KD + POODLE [53]	89.93	93.78
	EASY 4×ResNet12$\left(\frac{1}{2}\right)$ (ours)	90.50±0.19	93.50±0.09
36 M c	LR + DC [24]	79.56±0.87	90.67±0.35
	PT+MAP [56]	91.55±0.19	93.99±0.10
	iLPC [50]	91.03±0.63	94.11±0.30
	EASY 3×ResNet12 (ours)	90.56±0.19	93.79±0.10

**Table 8 jimaging-08-00179-t008:** The 1-shot and 5-shot accuracy of state-of-the-art methods and the proposed solution on **FC-100** in the **transductive** setting.

	Method	1-Shot	5-Shot
≤12 M c	TADAM [42]	40.10±0.40	56.10±0.40
	EASY 2×ResNet12$\left(\frac{1}{\sqrt{2}}\right)$ (ours)	54.47±0.24	65.82±0.19
36 M c	SIB + E${}^{3}$BM [55]	46.00	57.10
	fine-tuning (train) [54]	43.16±0.59	57.57±0.55
	ODC [48]	47.18±0.30	59.21±0.56
	fine-tuning (train+val) [54]	50.44±0.68	65.74±0.60
	EASY 3×ResNet12 (ours)	54.13±0.24	66.86±0.19

**Table 9 jimaging-08-00179-t009:** The 1-shot and 5-shot accuracy of state-of-the-art methods and the proposed solution on CIFAR-FS in the transductive setting.

	Method	1-Shot	5-Shot
≤12 M c	SSR [49]	76.80±0.60	83.70±0.40
	iLPC [50]	77.14±0.95	85.23±0.55
	DPGN [51]	77.90±0.50	90.02±0.40
	ECKPN [52]	79.20±0.40	91.00±0.50
	EASY 2×ResNet12$\left(\frac{1}{\sqrt{2}}\right)$ (ours)	86.99±0.21	90.20±0.15
36 M c	SSR [49]	81.60±0.60	86.00±0.40
	fine-tuning (train+val) [54]	78.36±0.70	87.54±0.49
	iLPC [50]	86.51±0.75	90.60±0.48
	PT+MAP [56]	87.69±0.23	90.68±0.15
	EASY 3×ResNet12 (ours)	87.16±0.21	90.47±0.15

**Table 10 jimaging-08-00179-t010:** The 1-shot and 5-shot accuracy of state-of-the-art methods and the proposed solution on TieredImageNet in the transductive setting.

	Method	1-Shot	5-Shot
≤12 M c	PT+MAP [56]	85.67±0.26	90.45±0.14
	TIM-GD [47]	79.90	88.50
	ODC [48]	83.73±0.36	90.46±0.46
	SSR [49]	81.20±0.60	85.70±0.40
	Rot + KD + POODLE [53]	79.67	86.96
	DPGN [51]	72.45±0.51	87.24±0.39
	EPNet [37]	76.53±0.87	87.32±0.64
	ECKPN [52]	73.59±0.45	88.13±0.28
	iLPC [50]	83.49±0.88	89.48±0.47
	ASY ResNet12 (ours)	83.98±0.24	89.26±0.14
36 M c	SIB + E${}^{3}$BM [55]	75.60	84.30
	SSR [49]	79.50±0.60	84.80±0.40
	fine-tuning (train+val) [54]	72.87±0.71	86.15±0.50
	TIM-GD [47]	82.10	89.80
	LR + DC [24]	78.19±0.25	89.90±0.41
	EPNet [37]	78.50±0.91	88.36±0.57
	ODC [48]	85.22±0.34	91.35±0.42
	iLPC [50]	88.50±0.75	92.46±0.42
	PEM${}_{n}$E-BMS* [38]	86.07±0.25	91.09±0.14
	EASY 3×ResNet12 (ours)	84.29±0.24	89.76±0.14

**Table 11 jimaging-08-00179-t011:** Ablation study of the steps of proposed solution in inductive setting, for a fixed number of trainable parameters in the considered backbones. When using ensembles, we use 2×ResNet12$\left(\frac{1}{\sqrt{2}}\right)$ instead of a single ResNet12.

Dataset	E	AS	1-Shot	5-Shot
MiniImageNet			68.43±0.19	83.78±0.13
		√	70.84±0.19	85.70±0.13
	√		68.69±0.20	84.84±0.13
	√	√	70.63±0.20	86.28±0.12
CUB-FS			74.13±0.20	89.08±0.11
		√	77.40±0.20	91.15±0.10
	√		75.01±0.20	89.38±0.11
	√	√	77.59±0.20	91.07±0.11
CIFAR-FS			73.38±0.21	87.42±0.15
		√	74.26±0.21	88.16±0.15
	√		74.36±0.21	87.82±0.15
	√	√	75.24±0.20	88.38±0.14
FC-100			45.68±0.19	62.78±0.19
		√	46.43±0.19	64.16±0.19
	√		47.52±0.19	63.92±0.19
	√	√	47.94±0.20	64.14±0.19
TieredImageNet			72.52±0.22	86.79±0.15
		√	74.17±0.22	87.81±0.14
	√		72.14±0.22	86.66±0.15
	√	√	73.36±0.22	87.37±0.15

**Table 12 jimaging-08-00179-t012:** Ablation study of the steps of proposed solution in **transductive** setting for a fixed number of trainable parameters in the considered backbones. When using ensembles, we use 2×ResNet12$\left(\frac{1}{\sqrt{2}}\right)$ instead of a single ResNet12.

Dataset	E	AS	1-Shot	5-Shot
MiniImageNet			80.42±0.23	86.72±0.13
		√	83.02±0.23	88.36±0.12
	√		80.27±0.23	87.45±0.12
	√	√	82.31±0.24	88.57±0.12
CUB-FS			86.93±0.21	91.53±0.11
		√	89.80±0.20	93.12±0.10
	√		87.28±0.21	91.89±0.10
	√	√	90.05±0.19	93.17±0.10
CIFAR-FS			84.18±0.23	89.56±0.15
		√	85.55±0.23	90.07±0.15
	√		84.89±0.22	89.60±0.15
	√	√	86.99±0.21	90.20±0.15
FC-100			51.74±0.23	65.39±0.19
		√	52.93±0.23	66.51±0.19
	√		53.39±0.23	65.71±0.19
	√	√	54.47±0.24	65.82±0.19
TieredImageNet			82.32±0.24	88.45±0.15
		√	83.98±0.24	89.26±0.14
	√		81.48±0.25	88.40±0.15
	√	√	83.20±0.25	88.92±0.14

## Data Availability

Available online: https://github.com/ybendou/easy (accessed on 14 June 2022).

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
