# Peer review of "Easy—Ensemble Augmented-Shot-Y-Shaped Learning: State-of-the-Art Few-Shot Classification with Simple Components"

_2313-433X, 2022, doi:10.3390/jimaging8070179_

Round 1

Reviewer 1 Report

The paper misses a clear goal definition.

The Conclusion is short and consequently cannot report the degree of goal fulfilment.

The paper only compares to other DL approaches. Reusing trained network substructures and just manipulating on feature vector and classifier level very much reminds on established domain-specific heuristic, possibly optimized, feature extraction methods followed by the same feature vector and classifier level  manipulation. I really miss a thorough comparison to such 'conventional' system design in effort and performance. In that sense, also references seem to lack completeness.

Further, it does not become really lucid in the paper, whether the focus of few-shot learning is on  establishing a 'wider scope' for existing classes in th recognition system and/or to introduce new classes ?

The issue of invariance seems only partially to be addressed, as, e.g., in the suggested cropping, images are 'chopped', but illumination, color change etc. seems not be included/covered by the approach.

How is such a method performing if it is confronted with freshly acquired data, e.g., camera images or videos, from the real world and a real application with all its variability instead on the 'stale' archive data ?

Reviewer 2 Report

Please add a reference for data thriftiness .

You are mentioning a repo, that is very good, but the repo contains the most important data in a Google Drive link. I would recommend moving  the : backbones, datasets and features in an open source environment.

When you fist mention "state of the art" in the introduction, I recommend to add 1-2 references to those methodologies and results. I know they are already referenced in the following pages , but the reader might want to have an idea on those things before reading the Methodology chapter.

I suggest to move Figure 1 from Introduction to Methodology.

Please add in the Conclusion a phrase about the qualitative and quantitative improvements you proposed in this paper.

Congratulations, this is a very good and relevant paper.

Reviewer 3 Report

This paper considers the creation of a benchmark bare-bones method for few-shot learning in a standard classification setting. The major contribution of the paper is to show that with proper training, a reasonably-sized model with few hyperparameters is sufficient for the classification task at hand - additional "bells and whistles" introduced elsewhere in the few-shot literature may not be as necessary as previously thought. Thank you for an interesting read.

I believe this paper presents some useful results, but is not publishable in its current form. I will provide comments on each section of the paper, with line numbers when possible/appropriate.

In general, the paper could be improved in terms of spelling (e.g., typos such as 'propsoed' in table captions) and grammar (e.g., the word "thereafter" is used when "hereafter" seems to be required, in several places). The services of a professional editor are recommended to improve the overall readability of the article.

Abstract:

Line 2: Is few-shot learning only applied in a classification setting? If there are other applications possible, then this definition of few-shot learning should be expanded.

Line 4 (and many other locations): the term "ingredients" is non-standard. I appreciate it may be difficult to find a suitable term here.

Lines 5-6: Does this mean: "If ensemble methods are built using models that are not optimally trained, a question then arises: is it better to use many weak predictors, or a single strong predictor?" The merits of using many weak predictors (an ensemble of weak models) have been examined in previous literature. At this point, as a general reader, I would be confused about the overall purpose of the paper. Once I read the whole thing through, it became clear that this was not what was meant, but I'm concerned that the citation count for this paper may be low if the purpose is not crystal clear in the abstract.

Line 9 (and elsewhere): the term "generic dataset" is not one that I've heard used frequently. From further reading, I believe "training dataset" was meant here, but I could be mistaken.

Introduction

Line 23: I also haven't heard of the term "data thriftiness". I think the general idea of this sentence was to emphasize that the full training set has many examples of all classes (except, of course, if some are held out for few-shot learning generalization purposes). Or perhaps "since the data set is large" would be enough?

Lines 24-25: "base" set - again, I haven't heard of this term, and perhaps readers of this journal would not have heard it either. I think "training" set is meant, but again, I could be mistaken.

Line 27: the word "fix" presents a bit of an issue here - it could mean either "repair" (in the sense of "update" for hyperparameters) or it could mean "hold constant". I think perhaps "update" or "optimize" would be a better word here.

Lines 27-29: A reference should be added here.

Lines 30-31: Drawing "knowledge" is perhaps not the clearest characterization of the process. I believe this is referring to the construction of feature vectors (i.e., feature extraction), which is probably not going to be seen by readers as meaning the same thing.

Lines 32-39: I believe this is meant to be a description of N-way-k-shot learning, but this term is not used and the description is unclear. Use of standard vocabulary would be very helpful for novice readers (e.g., new graduate students), as it gives a clear way to find more information.

Line 42: The term "backbone" is not one that I've seen generally applied in articles on few-shot learning. Do you have a reference for this term? If not, perhaps edit this sentence to say "which we term backbones" instead of "are generally termed backbones".

Lines 49-52: The terms inductive and transductive are usefully defined here, but a reference would be helpful.

Lines 54-56: Please provide some examples of the real-world applications mentioned here. If this section is more developed, it may lead to more citations from researchers attempting to find existing solutions to few-shot problems in industry.

Lines 62-63: Please provide evidence for the claim that previous methods were sub-optimally trained - if you can do so for the other methods against which you compare later in the paper, that would make the results presented much stronger, and much more likely to be cited. Since this is the main motivation of the paper, I don't think it can be published until such evidence is provided.

Lines 69-71: Here, it's stated that the goal is to provide a clear baseline. I think this is a laudable goal, as fair comparison can be difficult to achieve between models with differing numbers of hyperparameters (let alone parameters!), general structure, training method, etc. Please indicate what this baseline is - are you presenting a new baseline model, or a new baseline hyperparameter selection method, or...? I think this would make the paper much clearer to general readers and again allow the paper to be found more easily by those interested in citing it.

Figure 1 caption: "All the backbones are trained...also potentially different". This introduces a bit of confusion to the reader. The goal is to compare each model fairly, but then differences are introduced through initialization and batch order. This might be fine, if this is the general method used to create a baseline, but if so, a reference would be helpful. Right now, this sounds like it's not a fair comparison after all. Starting from different initializations can result in completely different parameter estimates after training is complete.

Related work

Line 100: "Most of the time..." implies that this is something that happens regularly. Please provide some references to strengthen this claim.

Lines 118-119: By "class probabilities distribution", I think "joint probability distribution over the classes" is meant?

Methodology

Lines 139-140: A new hyperparameter is introduced here, but I think one of the claims mentioned in the abstract was that the new method would not introduce additional hyperparameters?

Lines 174-176: Reducing the number of parameters may essentially cripple the ensemble methods, no? I'm not sure this achieves the goal of fair comparison. If this is a method used predominantly in the literature, some references would help here.

Lines 176-178: No results have been given regarding training time, yet it's claimed here that training time is "considerably reduced". If this is the case, some additional experiments to back up this claim would be beneficial, specifically comparing distillation to this method.

Results

Lines 194-196: A state-of-the-art (hereafter SOTA) claim is bold. If this truly is SOTA, I'm surprised that I didn't see it submitted to, say, NeurIPS or ICLR, but I may have missed it.

Lines 199-202: Please discuss why you think your method performs well in the case of imbalanced classes in the transductive setting. This could appear in a discussion after the presentation of the results.

Table 1

Was classification accuracy determined based on a count of the number of correctly classified images, divided by the total number of images? Accuracy, as described here, on its own, has been criticized as a method of evaluation. Have you considered using other metrics?

The FEAT method is missing an uncertainty.

What is the uncertainty being reported here? Standard deviation?

Table 2

Perhaps I am misunderstanding, but these results seem to imply the proposed method is not SOTA. The results for 1-shot learning overlap between the method in this study (EASY 2 x ResNet12(1/sqrt(2)) and invariance-equivariance). This is for FC-100 in the inductive setting, which on lines 195-196 was indicated as a SOTA result.

Table 5

Again, TieredImageNet in the inductive setting was referred to as SOTA previously, but, for example, results for the paper's method (ASY ResNet12) overlap with results from, say, DeepEMD v2. This doesn't look like SOTA to me.

Table 8

There is only one method (the one described in this paper) in the 12M setting. Could other methods be compared? Additional experiments may be required here, especially when a claim of SOTA is made (as has been done here). If there are no other experiments possible, something to this effect should be stated in the figure caption and a reason should be given.

Discussion (missing from the paper?)

In general, this paper would benefit from additional discussion. In cases where you achieve competitive results but not SOTA, what are the merits of your method compared to others? Why do you think your method falls short of SOTA in these situations? Other than the difference in parameters, are there other reasons why EASY may not be achieving similar results to the other methods in some cases? A more developed discussion would be helpful, that discusses the merits and (and detriments) of your method compared to the others against which you benchmarked. There are a few sentences in the first paragraph of the results presentation and the ablation study that start this off, but more discussion would be helpful to the reader.

Table 10

This table (as well as earlier tables) are missing some uncertainties.

Tables A1, A2, A3

All of these results are missing uncertainties; Table A1 is also missing the paper's model in the <=12M setting, I think.

References

Many of these are improperly formatted. I was unable to assess the validity of the references. For example, reference 18 has only the authors and a title in all-caps listed.

All in all, this paper has some interesting results, but could benefit from refinement. Thank you again for pursuing a baseline study, as I believe this could indeed be very useful for the few-shot learning community.

Reviewer 4 Report

The authors present their work regarding the problem of few-shot learning. A deep learning model namely Resnet12 is employed for initial learning on the generic dataset. That model which is called backbone being a feature extractor transfers learning to the few-shot procedure. That procedure consists of augmented sampling, backbone ensembling, vectors preprocessing and finally classification on a novel dataset. The novel dataset consists of a support dataset and a query dataset.  The methodology is proposed for the problem of inductive few-shot learning where data acquisition is expensive and the transductive few-shot problem where data labeling is expensive. A benchmark of the proposed methodology is provided on multiple classical datasets and  improved performance is shown.

The following issues should be addressed prior to publication:

  • In line 166 is mentioned that augmented feature vectors are generated from the validation and the novel dataset. In Fig. 1 only in the images of the novel dataset multiple crops are generated. Please clarify.
  • The ensembling is done by concatenating feature vectors obtained from different backbones through differentiation of seed. In line 174 is mentioned that when comparing a backbone with an ensemble of backbones the number of parameters in the ensemble backbones is reduced. Please explain it with a numerical example.
  • In section 3.4 (feature vectors preprocessing) is mentioned “Denote z the average feature vector of the base dataset if in inductive setting or of the few-shot considered problem if in transductive setting”. In the Introduction (lines 49 to 53) is mentioned that in inductive learning only the support dataset is available to the few-shot classifier and in transductive learning the few-shot classifier has access to both the support and the query datasets. The two statements don’t seem to match. Please clarify.
  • In tables 1 to 10 the results in terms of accuracy are presented for state-of-the-art models on different datasets. Since the configuration of the few-shot learning implies base, validation, support and query datasets please indicate the corresponding sizes for each case of dataset accordingly.

Round 2

Reviewer 3 Report

Thank you for addressing my comments. I realize they were quite extensive, and I appreciate the time taken. Below, I address the comments provided on my comments (yes, awkward phrasing, sorry!).

I now believe the abstract is a good characterization of the paper. Thank you for revising it.

There are still some typos (e.g., “hereaft”, which I don’t believe is a word), so some additional editing may be required. Also, I believe the term is N-way-k-shot, not n-ways-k-shot.

Thank you for clarifying training vs. generic vs. support data sets.

In lines 42-43 of the revised draft: “In this work, we do not consider the use of additional data such as other datasets, neither semantic nor segmentation information.” Is it typical in few-shot learning to incorporate additional datasets in the fashion of data augmentation? I’m not sure what this sentence is meant to convey.

In lines 54-56: “This is the case for FMRI data for example where it is difficult to generalize from one patient to another and collect hours of training data on a patient could be harmful.” – are there already some examples of few-shot learning in this setting? If so, citing a few of these papers would be helpful.

Lines 57-58: “Such situation can occur when experts must properly lable data ut the data itself is obtained cheaply.” It strikes me that this may happen very frequently in the medical field. One possible citation here might be
Henderson, Robert DE, et al. Automatic Detection and Classification of Multiple Catheters in Neonatal Radiographs with Deep Learning." Journal of Digital Imaging 34.4 (2021): 888-897 (in the interest of full disclosure: I know one of the authors, but I automatically thought of their work when I saw this discussion – please feel free to omit this reference if you don’t believe it to be appropriate).

Lines 64-66: “More problematically, we noticed that many of these contributions start with suboptimal training procedures or architectures.” – I realize Table 13 was added to try to address my previous comment (Comment 16), and I appreciate the attempt. I’m still not convinced this shows the reader that previous training methods were suboptimal. It shows that previous training methods exist, though. Perhaps a better point here would be that the method introduced in this paper is simpler (maybe in the sense of ease of training, or size, or interpretation, etc.) than other methods. That would be an easier point to make, and would still show the usefulness of your method. Indicating that previous methods were sub-optimally trained without providing clear evidence may be interpreted as un-collegial, but I can appreciate providing evidence on this point may also be problematic. Changing the point may be the best course of action.

On the response to Comment #18, yes, I did completely misunderstand originally. Thank you for providing this clarification. Now that I see the other backbones were not re-trained, I have further questions: Did you ensure the data splits were the same across all methods? Was identical methodology used for hyperparameter tuning? Was the experimental setup the same (e.g. number of epochs, etc.)? If the experiments were set up differently, then the goal of a fair comparison cannot be made.

The response to Comment #22 was an excellent point, and I think it’s much clearer now. Thank you!

On the response to Comment #29, “Regarding 1-shot learning on FC-100, our SOTA is on EASY 3xResNet12 with accuracy that is higher than those reported by invariance-equivariance. The results are indeed overlapping due to the high uncertainty reported by invariance-equivariace.” If this is the case, then claiming a SOTA result does not quite work. If the confidence intervals overlap, then there is no statistically significant difference between the results. Perhaps the argument to be made here lies in the lower uncertainty of your method compared to others such as invariance-equivariance?

On the new discussion section: I still do not see any speculation as to why this method outperforms others in the case of imbalanced classes in the transductive setting (comment #25). I see there’s a mention of the performance in the transductive setting, but no comment as to why it might achieve these better results. I think this is an important point because the transductive setting seems difficult, and your method does well. Explaining (or trying to explain!) why this may be would help the reader understand the merits of your method.
